# Bioguided Purification of Active Compounds from Leaves of *Anadenanthera colubrina* var. *cebil* (Griseb.) Altschul

**DOI:** 10.3390/biom9100590

**Published:** 2019-10-08

**Authors:** Daniel Rodrigo Cavalcante de Araújo, Túlio Diego da Silva, Wolfgang Harand, Claudia Sampaio de Andrade Lima, João Paulo Ferreira Neto, Bárbara de Azevedo Ramos, Tamiris Alves Rocha, Harley da Silva Alves, Rayane Sobrinho de Sousa, Ana Paula de Oliveira, Luís Cláudio Nascimento da Silva, Jackson Roberto Guedes da Silva Almeida, Márcia Vanusa da Silva, Maria Tereza dos Santos Correia

**Affiliations:** 1Instituto Nacional do Semiárido (INSA/MCTIC), 58429-500 Campina Grande, Brazil; wolfgang.harand@insa.gov.br; 2Departamento de Bioquímica, Centro de Biociências, Universidade Federal de Pernambuco (UFPE), 50670-901 Recife, Brazil; babi.a.ramos@gmail.com (B.d.A.R.); tamialvesinsl@gmail.com (T.A.R.); marciavanusa@yahoo.com.br (M.V.d.S.); mtscorreia@gmail.com (M.T.d.S.C.); 3Centro de Tecnologias Estratégicas do Nordeste (CETENE/MCTIC), 50670-901 Recife, Brazil; tulio.silva@cetene.gov.br; 4Departamento de Biofísica e Radiobiologia, Centro de Biociências, Universidade Federal de Pernambuco (UFPE), 50670-901 Recife, Brazil; claudia.salima@gmail.com (C.S.d.A.L.); joaopaulo.ferreiraneto@gmail.com (J.P.F.N.); 5Laboratório de Fitoquímica, Departamento de Farmácia e Pós-graduação em Ciências Farmacêuticas, Universidade Estadual da Paraíba (UEPB), 58429-500 Campina Grande, Brazil; harley.alves@hotmail.com; 6Laboratório de Patogenicidade Microbiana, Universidade CEUMA, 65080-805 São Luís, Brazil; rayane_sobrinho@hotmail.com; 7Central de Análise de Fármacos, Medicamentos e Alimentos, Universidade Federal do Vale do São Francisco (CAFMA/UNIVASF), 56328-903 Petrolina, Brazil; ana_tecquimica@yahoo.com.br (A.P.d.O.); jackson.guedes@univasf.edu.br (J.R.G.d.S.A.)

**Keywords:** *Staphylococcus aureus*, medicinal plant, compound purification

## Abstract

*Anadenanthera colubrina* var cebil (Griseb.) Altschul is a medicinal plant found throughout the Brazilian semi-arid area. This work performed a bioguided purification of active substances present in ethyl acetate extract from *A. colubrina* leaves. The anti-*Staphylococcus aureus* and antioxidant actions were used as markers of bioactivity. The extract was subjected to flash chromatography resulting in five fractions (F1, F2, F3, F4, and F5). The fractions F2 and F4 presented the highest antimicrobial action, with a dose able to inhibit 50% of bacteria growth (IN50) of 19.53 μg/mL for *S. aureus* UFPEDA 02; whereas F4 showed higher inhibitory action towards DPPH radical (2,2-diphenyl-1-picryl-hydrazyl-hydrate) [dose able to inhibit 50% of the radical (IC50) = 133 ± 9 μg/mL]. F2 and F4 were then subjected to preparative high-performance liquid chromatography (HPLC) and nuclear magnetic resonance (NMR), resulting in the identification of *p*-hydroxybenzoic acid and hyperoside as the major compounds in F2 and F4, respectively. Hyperoside and *p*-hydroxybenzoic acid presented IN50 values of 250 μg/mL and 500 μg/mL against *S. aureus* UFPEDA 02, respectively. However, the hyperoside had an IN50 of 62.5 μg/mL against *S. aureus* UFPEDA 705, a clinical isolate with multidrug resistant phenotype. Among the purified compounds, the proanthocyanidins obtained from F2 exhibited the higher antioxidant potentials. Taken together, these results highlight the potential of *A. colubrina* leaves as an alternative source of biomolecules of interest for the pharmaceutical, food, and cosmetic industries.

## 1. Introduction

The World Health Organization (WHO), in its document “Strategies on Traditional Medicine”, promotes the strengthening of quality assurance, safety, and proper use of medicinal plants. This document suggests the regulation of products and practices associated with medicinal plants [1]. This is important since the products derived from medicinal plants may not contain the active ingredient in adequate quantity due the environmental influences, absence of nutrients in the soil [2]. Other issues comprise chemical and microbial (or their toxins) contamination [3,4]. These failures contribute to the lack of confidence of medical professionals in prescribing their patients medicinal plants [5]. In this sense, although the population uses several products derived from plants (such as teas and tinctures), due to their therapeutic properties, the obtaining of purified compounds for high-precision identification is still necessary to characterize their composition [6,7]. For this, it is necessary to use organic solvents in extractive and chromatographic techniques that can be accompanied by chemical or biological tests, aiming at the isolation of the active compounds [8,9].

*Anadenanthera colubrina* var *cebil* (Griseb). Altschul is a plant from Fabaceae family that occurs in countries such as Brazil and Argentina [10,11]. *A. colubrina* stands out due its medicinal properties, which include the application of different tissues for treatment of inflammatory disorders [12,13]. In particular, *A. colubrina* is pointed as one of the most important medicinal plant of Caatinga area (the Brazilian semi-arid ecosystem), where it is popularly known as *angico* [14,15]. Several scientific studies have evidenced the pharmacological properties of products derived from *A. colubrina* (mostly extracts and fractions) that include antimicrobial [11,16,17], antioxidant [18,19,20,21], wound-healing [22,23], anti-inflammatory [24,25], antinociceptive [20,25], and antiproliferative actions [19,26]. Despite this therapeutic potential, little information is available about the chemical identity of the active(s) compound(s) responsible for each action. Therefore, the present work aimed to perform a bioguided study to obtain compounds from *A. colubrina* var *cebil* with antimicrobial and antioxidant activities.

## 2. Materials and Methods

### 2.1. Collection and Identification of the Specie Anadenanthera Colubrina Var Cebil

The aerial parts of *A. colubrina* var *cebil* were collected near to the area known as Pedra do Cachorro in the Catimbau Valley National Park (Buíque, Pernambuco, Brazil; the approximate coordinates are: 08°34′30.96″ S and 37°14′51.76″ O). The collection was performed in March 2015. The authors confirm that the authority designated Chico Mendes Institute for Biodiversity Conservation (ICMbio) granted permission through the System of Authorization and Information on Biodiversity (SISBIO) with authentication code n° 86962334. Exsiccates were prepared and the specimen was incorporated into the Dárdano de Andrade Lima herbarium of the Agronomic Institute of Pernambuco (IPA-PE; voucher protocol n° 80351).

### 2.2. Preparation of Extracts

The aerial parts were oven-dried at 45 °C (Marconi MA 035/511, Marconi Equipamentos Ltda, São Paulo, Brazil) for 3 days, and then ground in an industrial Blender (Skymsen Inox LB-25MB, Metalúrgica Skymsen Ltda, Santa Catarina, Brazil) until a powder was obtained. The powder was sieved (Mesh 35 and 14) and subjected to the Accelerated Solvent Extractor (ASE 350 Fisher Scientific, Pittsburgh, PA, USA) In the extractor, six extraction cells containing 20 g of the powder were subjected to extractions following the eluotropic series (hexane, ethyl acetate, and methanol). Each solvent was used for 45 min with a flow rate of 5 mL/min at 40 °C under a pressure of ±1500 psi. After this procedure, were obtained the hexane extract (ASE-Hex; 2.9% yield), ethyl acetate extract (ASE-AcOEt; 4.5%), and methanol extract (ASE-MeOH with 22.5%). The extract previously filtered in ASE 350 was concentrated in Rocket Evaporator (Fisher Scientific, Pittsburgh, PA, USA) at 40 °C.

### 2.3. Phytochemical Analysis and Purification

#### 2.3.1. Flash Chromatography

The ASE-AcOEt extract was submitted for fractionation using flash chromatography (Biotage Isolera one, Biotage company, Charlotte, NC, USA). The separation occurred on the SNAP KP-SIL (Biotage company, Charlotte, NC, USA) 50 g column through a mobile phase with a gradient of toluene:ethyl acetate (PhME: AcOEt, 20:70) with 1 column volume (1 CV), followed by 100% AcOEt (6 CV) and then for AcOEt: 70:20 MeOH (6 CV) with flow rate of 70 mL/min and scanning detection of 200–800 nm. The fractions were grouped by software that suggested the fractions according to the UV absorption spectrum, and thus five fractions were grouped: F1, F2, F3, F4, and F5.

#### 2.3.2. High-Performance Liquid Chromatography (HPLC-DAD)

The extracts, fractions, and isolated compounds were analyzed by HPLC (1260 infinity LC System-DAD, Agilent OpenLAB CDS EZChrom Edition software, version 04.05 of Agilent Technologies, Santa Clara, CA, USA). The samples were prepared to obtain a concentration of 5 mg/mL and were sonicated until complete solubilization. They were filtered with 0.22 µm PTFE filters and then analyzed by HPLC. The compounds separation occurred via the column: Zorbax, SB-C18, 5 µm and 4.6 × 250 mm with 5 µm Zorbax SB-C18 pre-column and 4.6 × 12.5 mm. The solvents used were: Mili-Q water (Millipore, Burlington, MA, USA) with 0.3% acetic acid (A) and acetonitrile (B) (LiChrosolv, Merck, Darmstadt, Hessen, Germany) following a linear gradient of 92–65% (A) 0–15 min; the initial conditions returning 65–92% (A), then at 40 °C, flow rate 2.4 mL/min, initial pressure 210 bar, and scanning detection at 190–400 nm. This method was accomplished after an exploratory method of 60 min and flow of 1 mL/min and was considered as satisfactory for analyses of the samples’ contents and for the standards gallic acid, *p*-coumaric acid, caffeic acid, catechin, ferulic acid, luteolin, isoquercetin, chlorogenic acid, quercetin, rutin, ellagic acid, and geraniin.

#### 2.3.3. Preparative High-Performance Liquid Chromatography Coupled To Mass Spectrometry (HPLC-MS)

The fractions F2 and F3 from the flash chromatography system were analyzed using the preparative HPLC (Waters autopurification system, Milford, MA, USA) with fraction collector, UV detector 2489, and ACQUITY QDa. The fractions were dissolved in MeOH and filtered with 0.22 µm PTFE filters until several samples with a final value of 15 mg/mL were obtained. The separation and isolation of the compounds occurred via the preparative column XBridge Prep C18 (5 µm and 10 × 100 mm), by means of the following mobile phase; Mili-Q water with 0.1% formic acid (A) and acetonitrile (B) (LiChrosolv, Merck, Darmstadt, Hessen, Germany) with linear gradient 94–65% (A) 0–9 min; the initial conditions were then returned to room temperature, flow rate of 9 mL/min, and detection at 256 nm; only the main and unknown constituents were collected.

#### 2.3.4. Ultra-High-Performance Liquid Chromatography Coupled to Mass Spectrometry (UPLC-MS)

The ASE-AcOEt extract was analyzed in AQUITY H-Class (Milford, MA, USA) on a BEH 2.1 × 100 mm, 1.7 µm column. The mobile phase consisted of aqueous solution containing 2% MeOH, 5 mM ammonium formate and 0.1% formic acid (A) and methanolic solution containing 0.1% formic acid (B), which were pumped into a flow rate of 0.3 mL/min, and 10 µL of the sample were injected. The elution was performed in gradient mode and the initial condition 98% of the eluent A was maintained for 15 s. The proportion of the phase B was linearly increased to 99% in 8.5 min, remaining at 99% for 1 min, then returning to the initial conditions of analyses. The column oven was maintained at 40 °C. The data acquisition was done in the full-scan mode, searching for masses between 100 and 1000 Da, in negative ionization. The acquisition of the chromatograms and mass spectra was done through MassLynx software.

#### 2.3.5. Nuclear Magnetic Resonance (NMR)

The compounds isolated by preparative HPLC-MS were analyzed by NMR. 1D-NMR (^1^H and ^13^C (1H)) and 2D (^1^H-^1^H COSY, ^1^H-^13^C HSQC, and ^1^H-^13^C HMBC) were acquired at 23 °C in deuterated dimethyl sulfoxide (DMSO-d^6^) in a Bruker ASCEND III 400 a 9.4 T (Billerica, MA, USA), observing ^1^H and ^13^C at 400 and 100 MHz, respectively. The NMR spectrometer was equipped with a 5 mm multinuclear direct detection probe (BBO probe) with gradient z. The correlation experiments of ^1^H-^13^C one-link (HSQC) and long-range (HMBC) were optimized for the mean coupling constant 1 J (C, H) and LRJ (C, H) of 140 and 8 Hz, respectively. All ^1^H and ^13^C-NMR (δ) NMR chemical shifts are given in ppm relative to the TMS signal at 0.00 ppm as an internal reference, and the coupling constants (J) in Hz.

### 2.4. Antimicrobial Activity

#### 2.4.1. Microbial Strains

The antimicrobial assay was performed with *Staphylococcus aureus* UFPEDA 02 (=ATCC 6538) and *S. aureus* UFPEDA 705, both provided by the Culture Collection of the Department of Antibiotics from Federal University of Pernambuco (UFPEDA). *S. aureus* UFPEDA 705 is a methicillin-resistant isolate (Methicillin-Resistant *S. aureus* (MRSA)) obtained from surgical wound, and resistant to diverse antibiotics, including drugs from the beta-lactam group (e.g., ampicillin, oxacillin, cephalothin, cefoxitin, cefepime, and cefuroxime), nalidixic acid, nitrofurantoin, and gentamicin [27].

#### 2.4.2. Antimicrobial Assay

The antimicrobial action was performed using a broth microdilution assay based on the detection of bacterial growth by spectrophotometric measurement, as proposed by Quave et al. [28]. The ASE-AcOEt extract, fractions (F1, F2, F3, F4, and F5) and the compounds (hyperoside, *p*-hydroxybenzoic acid and proanthocyanidins) were dissolved in 10% aqueous solution of DMSO until a homogeneous mixture was obtained (5000 μg/mL for extract and fraction; 1000 μg/mL for isolated compounds). In a 96-well microplate, serial dilutions of each sample were prepared in Muller–Hilton Broth (MHB). The concentrations tested for ASE-AcOEt extract and its fractions (F1, F2, F3, F4, and F5) ranged from 2500 to 9.76 μg/mL; whereas purified compounds were tested from 500 μg/mL to 1.9 μg/mL. Following, each well received 20 µL of bacterial suspension (approximately 1.5 × 10^8^ CFU/mL). Untreated bacteria and vehicle-treated bacteria were used as positive controls of microbial growth. The plates were read in a spectrophotometer at 600 nm after the serial dilution (T0h) and after 24 h (T24h) of oven conditioning at 37 °C. Subsequently, the IN50 (concentration able to inhibit 50% of bacterial growth), was calculated according to the equation below [28].

Equation: % inhibition = [1 − (ODT24h − ODT0h)/(ODgc24h − ODgc0h)] × 100
(1)
where ODT24h = optical density (600 nm) of the test plate at 24 h of inoculation; ODT0h = optical density (600 nm) of the test plate shortly after inoculation; ODgc24h = optical density (600 nm) of the bacterial growth control wells after 24 h of inoculation; and ODgc0h = optical density (600 nm) of the bacterial growth control wells shortly after inoculation.

### 2.5. Antioxidant Activity

#### 2.5.1. Free Radical Sequestration

In this assay, the free radical scavenging activity of each sample (ASE-AcOEt extract, fractions (F1, F2, F3, F4, and F5), and the compounds (hyperoside, *p*-hydroxybenzoic acid, and proanthocyanidins)) was measured in terms of hydrogen donation using 2,2-diphenyl-1-picrihydrazyl radical (DPPH) [29]. An aliquot of 250 µL of DPPH methanolic solution (with absorbance of 0.7 at 517 nm) was mixed with 40 µL of different sample concentrations which were previously dissolved in methanol (31.25, 62.5, 125, 500, and 1000 μg/mL). After 25 min, the absorbance at 517 nm was measured. Gallic acid was used as the reference compound and the negative control was the DPPH solution added to 40 µL of methanol. The elimination of DPPH radicals was calculated by the formula

Inhibition of DPPH (%) = [(Aa − Ac)/Ac] × 100
(2)
where Aa is absorbance of the sample and Ac is control absorbance. The concentration that inhibited 50% (IC50) of DPPH radical was calculated by linear regression.

#### 2.5.2. Total Antioxidant Activity (TAA)

The purified compounds (hyperoside, *p*-hydroxybencoic acid, and proanthocyanidins) were dissolved to the concentration of 1 mg/mL in methanol, and 0.1 mL of each sample was mixed with 1 mL of the phosphomolybdenum solution (600 mM sulfuric acid, 28 mM sodium phosphate, and 4 mM ammonium molybdate), and then incubate in water at 95 °C for 90 min. After returning to room temperature, the absorbance of each samples were measured at 695 nm against a blank (1 mL of solution and 0.1 mL of methanol) [30]. The total antioxidant activity was expressed in relation to ascorbic acid and calculated by the formula

TAA (%) = [(Aa − Ac)/(Aaa − Ac)] × 100
(3)
where Ac is control absorbance, Aa is sample absorbance, and Aaa is absorbance of ascorbic acid.

#### 2.5.3. Reduction of Ferric ion (FRAP Method)

The FRAP assay was performed with the compounds: hyperoside, *p*-hydroxybenzoic acid, and purified proanthocyanidins. The assay was done according to Benzie et al. [31] with modifications. The FRAP reagent was freshly prepared by mixing stock solution of 300 mM acetate buffer (pH 3.6), 10 mM 2,4,6-tripyridyl-s-triazine (TPTZ) (solubilized in 40 mM HCl), and 20 mM FeCl_3_ in a proportion 1:1:10 (*v*/*v*/*v*). Samples of 0.07 mL of the purified compounds were used in a concentration of 1 mg/mL and mixed with 0.2 mL of the FRAP reagent and allowed to stand for 30 min at 37 °C in the dark. Subsequently the samples were read at 593 nm. A standard curve was made with FeSO_4_ (0–1000 μg/mL) (y = 0.0054x + 0.1465 and R^2^ = 0.9874). The results are expressed in μg FeSO_4_ (II)/mg and compared with gallic acid under the same conditions.

### 2.6. Statistical Analysis

All assays were performed in triplicate in at least two independent experiments. The results were expressed as mean and standard deviation. The data were submitted to analysis of variance (ANOVA) and for comparisons between the means the Tukey test was used through the software’ Past version 2.17. The differences were considered statistically significant when *p* < 0.05.

## 3. Results

### 3.1. Purification of Compounds from Anadenanthera Colubrina 

The qualitative analysis by HPLC-DAD of ASE-AcOEt extract indicated the presence of quercetin and low levels of catechin, *p*-coumaric acid and gallic acid. The acquisition of the 3D chromatogram with scanning from 190 to 400 nm (Figure 1) pointed the presence of approximately 10 compounds with absorption greater than 200 mAU. Among the major compounds, specifically between the retention times (Rt) 4 and 6 min, is *p*-hydroxybencoic acid (*m*/*z* of 137 [M + H]), which was purified by preparative HPLC and identified by NMR. Between the retention times of 8 and 10 min, two signals with UV spectra characteristic of quercetin derivatives were detected (Figure 1). One of them was the hyperoside compound (*m*/*z* of 487 [M + Na]^+^), which was also purified and identified as previously described. This is the first report of purification and identification of hyperoside in aerial parts of *A. colubrina*, this compound represents approximately 15% of the whole sample. At the retention times of 10 and 12 min, two proanthocyanidins (*m*/*z* of 530 [M]) were identified based on analyzes of mass spectrometry.

### 3.2. Bioguided Purification of Compounds from Anadenanthera Colubrina

The ASE-AcOEt extract had IN50 values of 312.5 μg/mL and 2500 μg/mL against *S. aureus* UFPEDA 02 and *S. aureus* UFPEDA 705, respectively. In relation to antioxidant activity, the extract showed an IC50 142 ± 10 μg/mL in DPPH assay. This extract was submitted to flash chromatography, which generated 35 fractions that were grouped according to the UV absorption profile in five groups: F1, F2, F3, F4, and F5 with yields of 7, 16, 16, 42, and 7.6%, respectively (Figure 2). The fractions F2 and F4 were the most promising as antioxidants (IC50 of 202 ± 6 μg/mL and 133 ± 9 μg/mL in DPPH assay, respectively) and antimicrobials agents (IN50 of 19.53 μg/mL for both samples towards *S. aureus* UFPEDA 02) (Table 1). These two fractions proceeded to the preparative HPLC system.

The major compound in F2 had maximum absorption at 255 nm and it was isolated with high purity and identified by NMR as *p*-hydroxybencoic acid (**1**) (Figure 2). In the same fraction, two compounds that appeared to be proanthocyanidins (**2** and **3**) (531.44 *m*/*z* ES+) were also isolated. Compounds derived from quercetin were detected in high levels in F4, being one of them identified as hyperoside (**4**) (Figure 2). These results were confirmed by UPLC-MS.

### 3.3. Nuclear Magnetic Resonance–NMR

*p*-hydroxybenzoic acid compound (**1**) was obtained as a beige amorphous solid; UV (solvent) λmax 255 nm; NMR of ^1^H (MeOD-*d_4_*, 400 MHz) δH 7.87 (2H, d, *J* = 8.8 Hz, H-2 and H-6), 6.87 (2H, d, *J* = 8.8 Hz, H-3 e H-5); RMN 1^3^C (MeOD-*d_4_*, 100 MHz), δ_C_ 170.56 (-COOH), 163.38 (C-4), 133.1 (C-2 e C-6), 123.2 (C-1), 116.1 (C-3 e C- 5). In the ^1^H-NMR spectrum, the presence of a pair of doublets with constant coupling constant of 8.8 Hz (7.87 and 6.87), characteristic of the substituted 1–4 aromatic rings, was observed [32].

The ^13^C-NMR spectrum shows five carbon signals where this profile, and the hydrogen signals, confirmed the suggestion of a substituted 1–4 aromatic compound. The signal at 170.56 ppm was attributed to the carbon of the carboxylic acid group, and the signal at 163.3 ppm was attributed to an oxygenated aromatic carbon (C-4). The groups’ positions were confirmed by the analysis of the two-dimensional spectrum, in which the homonuclear correlation ^1^H-^1^H COSY showed us a single correlation between the signals 7.87 and 6.87. The heteronuclear correlation of ^1^H × ^13^C-HMBC multiple bonds showed the correlations among signal at 7.87 (H-2 and H-6) and signals 170.56 (-COOH, *J*^3^), 163.38 (C-4, *J*^3^)116.11 (C3 and C5, *J*^2^) and correlations among signal 6.87 (H-3 and H-5) and signals 163.38 (C-4, *J*^2^) and 123.23 (C-1, *J*^3^). Data analysis and comparison with the literature allowed the compound to be identified as *p*-hydroxybenzoic acid [33].

Quercetin 3-*O*-β-galactopyranose or hyperoside (compound **4**) HMRS: ^1^H-NMR (DMSO-*d*_6_, 400 MHz): 3.29 (1H, H-5′′), 3.34 (1H, H-4′′), 3.36 E 3.45 (2H, H-6′′), 3.60 (1H, H-2′′), 3.66 (1H, H-3′′), 5.36 (1H, H-1′′), 6.21 (1H, H-8), H-6), 6.81 (1H, d, *J* = 8.8 Hz, H-5), 7.47 (1H, H-2′), 7.59 (1H, d, *J* = 8.8 Hz, H-6′), 12.64 (1H, OH-5). The structure of hyperoside was identified by analyzing key signals from 1D and 2D-NMR spectroscopic data and comparison with the literature. The ^1^H-NMR spectrum showed a characteristic δ_H_ 12.64 signal of a C-5 hydroxyl group chelated on a structural flavonol and singlets at δ_H_ 6.21 and 6.43 typical of the meta-substituted ring A. Distorred doublets at δ_H_ 6.81 and 7.59 with coupling constant at 8.8 Hz and a singlet at δ_H_ 7.47 were observed, suggesting the presence of a quercetin derivative. The anomeric hydrogen of unit sugar, δ_H_ 5.37, showed direct coupling (^1^H × ^13^C in HSQC experiment) among δ_H_ signals 5.37 and 101.5 typical of the galactose structure and had its position based on literature data [34].

### 3.4. Antimicrobial Activity of Isolated Compounds

The antimicrobial activity of each compound [*p*-hydroxybenzoic acid (**1**), proanthocyanidins (**2**) and (**3**), and hyperoside (**4**)] isolated from the most active fractions (F2 and F4) was tested against *S. aureus* strains, however the IN50 values obtained were higher than those obtained from the fractions. The proanthocyanidin (**3**) showed the lowest IN50 value against *S. aureus* UFPEDA 02 (IN50 = 62.5 μg/mL, but it was not effective towards *S. aureus* UFPEDA 705 (IN50 > 500 μg/mL). The hyperoside (**4**) exhibited IN50 values of 250 μg/mL and 62.5 μg/mL against *S. aureus* UFPEDA 02 and *S. aureus* UFPEDA 705, respectively. The other two compounds showed IN50 values ≥ 500 μg/mL (Table 1).

### 3.5. Antioxidant Action of Isolated Compounds

As the fractions F2 and F4 were the most active in DPPH assay, we also evaluated the activity of their purified compounds using DPPH, TAA, and FRAP methods. The *p*-hydroxybenzoic compound was not able to inhibit the DPPH radical, and proanthocyanidins were the most antioxidant components in the F2 fraction, with IC50 of 124.8 ± 7 μg/mL and 178.4 ± 5 μg/mL for proanthocyanidins (**2**) and (**3**), respectively. Proanthocyanidins were also more active in total antioxidant activity (TAA) with 60 ± 3.3% (**2**) and 35 ± 0.67% (**3**). On the other hand, the main component of the F4 fraction (hyperoside; compound (**4**) had an IC50 of 258 ± 20 μg/mL in DPPH assay and a weak TAA (4.31 ± 0.81%). Finally, the compounds had similar results in FRAP method with values of 529.91 ± 21.55 μg FeSO_4_/mg, 550.22 ± 10.43 μg FeSO_4_/mg, and 552.62 ± 5.23 μg FeSO_4_/mg for hyperoside (**4**) and proanthocyanidins (**2**) and (**3**), respectively (Table 1).

## 4. Discussion

In the present work, we reported the bioguided purification and identification of compounds present in the leaves of *A. columbrina*, a medicinal plant found in Brazil [12,13]. The bioguided approach was performed using the anti-*S. aureus* and antioxidant actions as markers. Previous reports obtained by our group showed the antimicrobial potential of hydroalcoholic extracts from leaves of *A. columbrina*, in special against *S. aureus*. In brief, these data indicated that the ethyl acetate extracts (obtained by liquid–liquid extraction or by Soxhlet) have higher activity towards Gram-positive bacteria, such as *S. aureus* [35,36]. In addition, antioxidant activity was chosen as marker due the involvement of oxidative stress in the pathogenesis of diverse clinical situations that include cancer, sepsis, diabetes and neurogenerative diseases [37,38,39]. 

*S. aureus* are found in the skin, nasal cavity, and mucosal surfaces of at least one-third of the human population, these organisms can cause infections in the skin, lungs, joints, blood and central nervous system [40,41,42,43,44]. In particular, *S. aureus* are prevalent in the hospital environment and associated with many cases of prolonged hospitalizations and deaths [41,42,45]. Several strains of this species have also acquired resistance to multiple drugs and hypervirulent profiles; these characteristics make the treatment options for *S. aureus* very limited [46,47]. These features make *S. aureus* is one of the most devastating pathogens and highlight the need to identify new effective compounds [48,49]. In this way, the search for active molecules with therapeutic potential is of great importance.

Based on the anti-*S. aureus* activity, two fractions were selected for purification, leading to four isolated compounds: *p*-hydroxybenzoic acid (**1**), two proanthocyanidins (**2** and **3**), and hyperoside (**4**). These compounds were also detected as major constituents in the hydroalcoholic extract (data not shown). Proanthocyanidins and *p*-hydroxybenzoic acid were the major components of the F2, whereas the hyperoside and other major quercetin derivatives were the main molecules found in F4. In addition, we observed that those fractions with the lowest levels of these compounds showed the higher IN50 values (F1 and F5 with IN50 values > 500 μg/mL).

The hyperoside showed the higher antimicrobial potential (IN50 ≤ 250 μg/mL for both used strains) among the compounds obtained from leaves of *A. columbrina*. Similar results were recently reported by Ren et al. [50], which showed that hyperoside had a minimal bacteridal concentration (MIC) of 250 μg/mL against *S. aureus* ATCC 25923. Although hyperoside has been found as a component of some plants with antimicrobial and antioxidant activity [50,51,52,53,54], its mechanism of antimicrobial action of has not been reported yet. In our work, hyperoside showed the lowest in vitro antioxidant action among the tested compounds; however, this biomolecule has shown antioxidant properties in in vivo [55,56,57] and cell-based models [56,58,59]. The antioxidant activity of hyperoside is also highlighted by its anti-inflammatory effects as shown by several reports [54,60,61,62]. Other promising pharmacological properties include the treatment of cancer [63,64], obesity [65], arthritis [61], and diabetes [66].

The other isolated biomolecule obtained from *A. columbrina* was *p*-hydroxybenzoic acid, which showed activity against *S. aureus* UFPEDA02; however, it did not show inhibitory action towards the DPPH radical. The weak antiradical activity of *p*-hydroxybenzoic acid was observed by other authors [67]. In contrast, this compound has been pointed as antimicrobial agent against *S. aureus*, *Staphylococcus epidermis*, *Escherichia coli*, *Pseudomonas aeruginosa*, and *Salmonella typhimurium* [33,68,69], an effect usually associated with its relative hydrophobicity [70]. The reported MIC values against *S. aureus* ranged from 3 to 926 μg/mL [33,69], these differences may be related to the strains used in each study and methods applied. *p*-Hydroxybenzoic acid is also reported to have potential for the treatment of diabetes [71,72], neurodegeneration [73], and inflammatory disorders [74,75].

The proanthocyanidins purified from *A. colubrina* showed activity only against the standard *S. aureus* strain, but they were not effective against the clinical isolates. However, some reports highlight the effectiveness of these class as inhibitors of *S. aureus* growth [76,77]. On the other hand, these proanthocyanidins from *A. colubrina* showed the highest antioxidant properties. These compounds are indicated as useful for treatment of various diseases related to oxidative stress such as inflammatory, neurodegenerative, and cardiovascular disorders [78,79,80].

## 5. Conclusions

The bioguided approach (based on antioxidant and antimicrobial actions) employed in this study resulted in the purification of four bioactive compounds leaves of *A. colubrina*: two proanthocyanidins, *p*-hydroxybenzoic, and hyperoside. These compounds have been described as potential alternatives for the treatment of several diseases, such as diabetes, arthritis, and other inflammatory disorders. Herein, hyperoside is the biomolecule with higher anti- *S. aureus* activity; however, the compounds showed lower activity than the semipurified fractions. These data suggest that the compounds should act in combination to provide bacterial inhibition. This work also showed that the proanthocyanidins exhibited the highest in vitro antioxidant activity among the molecules purified from *A. colubrina*. 

Although these compounds have been described as potential antimicrobial agents, the action mechanisms involved in the inhibition of *S. aureus* still need to be addressed in another study. As these compounds exhibited antioxidant and antimicrobial actions, they anti-infective properties could also be examined in future researches. Taken together, these results highlight the potential of *A. colubrina* leaves as an alternative source of bioactive compounds of interest for the pharmaceutical, food, and cosmetic industries.

## Figures and Tables

**Figure 1 biomolecules-09-00590-f001:**
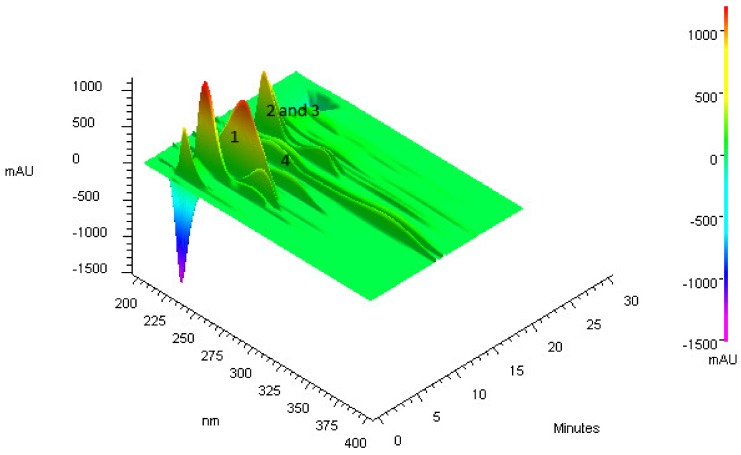
3D chromatogram of an exploratory analysis of the ASE-AcOEt extract. In the image we can see approximately 10 major compounds. Compound **1** (λmax 255 and *m*/*z* 137 [M + H]^−^) is *p*-hydroxybenzoic acid; compound **2** (λmax 280) and compound **3** are (λmax 281) proanthocyanidins; and compound **4** is (λmax 256 and 355 and *m*/*z* 487 [M + Na]^+^) hyperoside.

**Figure 2 biomolecules-09-00590-f002:**
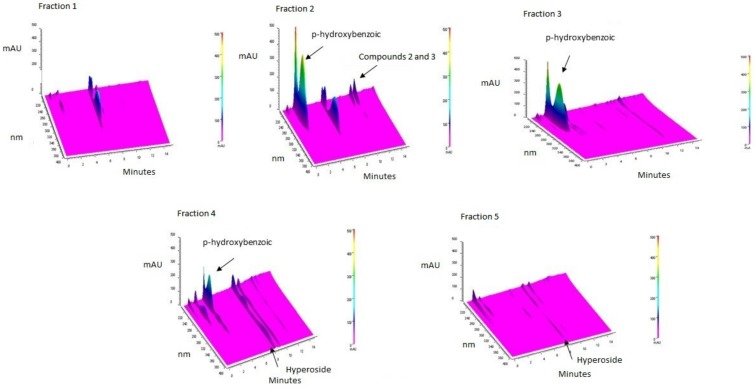
3D chromatogram of fractions F1, F2, F3, F4, and F5. The chromatogram 3D allows visualizing the compounds by means of their respective UV spectra (**1**: *p*-hydroxybenzoic acid, **2** and **3**: proanthocyanidins and **4**: hyperoside), their position in the chromatogram with respect to time on a scale of 190 to 400 nm. mAU: Signal strength; nm: wavelength; t: time.

**Table 1 biomolecules-09-00590-t001:** Antimicrobial and antioxidant assay of the extract, fractions and compounds isolated from *Anadenanthera colubrina* var *cebil*.

Samples	Antimicrobial Activity		Antioxidant Activity		
	IN50 (μg/mL)		DPPH	TAA	FRAP
	*S. aureus* UFPEDA 02	*S. aureus* UFPEDA 705	IC50 (μg/mL)	%	μg FeSO_4_/mg
ASE-AcOEt	312.5	2500	142 ± 10	N/E	N/E
Fraction 1 (F1)	2500	2500	N/A	N/E	N/E
Fraction 2 (F2)	19.53	2500	202 ± 6 ^a^	N/E	N/E
Fraction 3 (F3)	78.12	>2500	263 ± 17 ^b^	N/E	N/E
Fraction 4 (F4)	19.53	>2500	133 ± 9 ^c^	N/E	N/E
Fraction 5 (F5)	>2500	>2500	218 ±7.6 ^a^	N/E	N/E
*p*-hydroxybenzoic acid (**1**)	500	>500	N/A	N/E	N/E
Proanthocyanidin (**2**)	500	>500	124.8 ± 7 ^c^	60 ± 3.3 ^a^	550.22 ± 10.43
Proanthocyanidin (**3**)	62.5	>500	178.4 ± 5 ^a^	35 ± 0.67 ^b^	552.62 ± 5.23
Hyperoside (**4**)	250	62.5	258 ± 20 ^b^	4.31 ± 0.81 ^c^	529.91 ± 21.55
Ampicillin	<1.9	3.9	N/E	N/E	N/E
Tetracycline	<1.9	15.6	N/E	N/E	N/E
Trolox	N/E	N/E	N/E	N/E	566.08 ± 13.81
Gallic acid	N/E	N/E	<312	N/E	N/E
Ascorbic acid	N/E	N/E	N/E	100	N/E

Legend: N/A: no action; N/E: not evaluated. In each column the values with significant differences (*p* < 0.05) are indicated by different letters ^a,b,c^.

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
