# Peer review of "Bioguided Purification of Active Compounds from Leaves of Anadenanthera colubrina var. cebil (Griseb.) Altschul"

_biomolecules, 2019, doi:10.3390/biom9100590_

Round 1
Reviewer 1 Report
Dear Authors,
The manuscript ID: biomolecules-593126-v1 entitled “Bioguided purification of active compounds from leaves of Anadenanthera colubrina var. cebil (Griseb.) Altschul” written by Daniel Rodrigo Cavalcante de Araújo, Túlio Diego da Silva, Wolfgang Harand, Claudia Sampaio de Andrade Lima, João Paulo Ferreira Neto, Bárbara de Azevedo Ramos, Tamiris Alves Rocha, Harley da Silva Alves, Rayane Sobrinho de Sousa, Ana Paula de Oliveira, Luís Cláudio Nascimento da Silva, Jackson Roberto Guedes da Silva Almeida, Márcia Vanusa da Silva and Maria Tereza dos Santos Correia is an interesting paper. The authors presented the potential of A. colubrina leaves as an alternative source of interesting bioactive compounds for some industries.
However, I have some objections to this article:
The antimicrobial assay of the extract and compounds (from leaves of Anadenanthera colubrina cebil (Griseb.) Altschul and control antibiotics) was not performed according to European Committee on Antimicrobial Susceptibility Testing (EUCAST) [EUCAST 2003] guidelines against reference strains of microorganisms from American Type Culture Collection (ATCC). The research methodology regarding the minimum inhibitory concentration (MIC) assessment is unclear and inappropriate.Reference:
[EUCAST 2003] European Committee for Antimicrobial Susceptibility Testing (EUCAST) determination of minimum inhibitory concentrations (MICs) of antibacterial agents by broth dilution (2003) EUCAST discussion document E. Dis 5.1. Clin Microbiol Infect 9: 1-7.
In section 2. Materials and Methods, Lines 143-144: “The antimicrobial assay was performed with Staphylococcus aureus ATCC 02 and aureus UFPEDA 705”. Further on in the text are strains: S. aureus UFPEDA 02 and S. aureus UFPEDA 705. Please harmonize this data. The preliminary antibacterial activity should be performed on reference strains and then on several wild or clinical isolates. In addition, only two strains of S. aureus were used for these studies. Tests should be also extended to other microorganisms. The publication would be more valuable and reliable with these additional results. Antimicrobial analyses are incomplete: there is no information about MBC (Minimal Bactericidal Concentration) and MBC/MIC ratios of extract and compounds.
Other minor mistakes:
Line 47: compounds [8,9] – compounds [8,9].
Line 53: known as angico [14,15] – known as angico [14,15].
Line 81: 40° C. – 40°C.
In the all text: “proanthcynidins” – proanthocyanidins
Line 231: Purification – purification
Line 238: in in DPPH assay – in DPPH assay
Page 8, Table 1: Antimicrobian Activity – Antimicrobial Activity
Line 287: p-hydrobromic acid (1) – p-hydroxybenzoic acid (1)
With highest regards,
Author Response
Dear reviewer,
We would like to thank you for all suggestions that improved the overall quality of our manuscript. Following we provide detailed answers for each point.
Reviewer comment: The antimicrobial assay of the extract and compounds (from leaves of Anadenanthera colubrina cebil (Griseb.) Altschul and control antibiotics) was not performed according to European Committee on Antimicrobial Susceptibility Testing (EUCAST) [EUCAST 2003] guidelines against reference strains of microorganisms from American Type Culture Collection (ATCC). The research methodology regarding the minimum inhibitory concentration (MIC) assessment is unclear and inappropriate.
Our response: Dear reviewer, we understand your concern. We determined the antimicrobial action of each sample by determine the percentage of inhibition, as reported for Quave et al. (2008). We made clearer in the text that we did not used the standard protocol used for commercially available drugs. We also changed the nomenclature for MIC50.
Quave, C. L.; Plano, L. R.; Pantuso, T.; Bennett, B. C., Effects of extracts from Italian medicinal plants on planktonic growth, biofilm formation and adherence of methicillin-resistant Staphylococcus aureus. J Ethnopharmacol 2008, 118, 418-28.
Reviewer comment: In section 2. Materials and Methods, Lines 143-144: “The antimicrobial assay was performed with Staphylococcus aureus ATCC 02 and aureus UFPEDA 705”. Further on in the text are strains: S. aureus UFPEDA 02 and S. aureus UFPEDA 705. Please harmonize this data. The preliminary antibacterial activity should be performed on reference strains and then on several wild or clinical isolates. In addition, only two strains of S. aureus were used for these studies. Tests should be also extended to other microorganisms.
Our response: We corrected the name of the strain in the text. In addition, it is important to highlight that the strain UFPEDA 02 is internal code for ATCC 6538, the standard strain used in evaluation of antimicrobial compounds. The strains UFPEDA 705 is a clinical isolate with multidrug resistance phenotype.
Reviewer comment: The publication would be more valuable and reliable with these additional results. Antimicrobial analyses are incomplete: there is no information about MBC (Minimal Bactericidal Concentration) and MBC/MIC ratios of extract and compounds.
Our response: We understand the reviewer suggestion, however we would like to highlight that the main goal of our study was the purification of bioactive molecules from leaves of Anadenanthera colubrina. For this we employed a bioguided approach using the anti-S. aureus action (previously reported by our group) and antioxidant activity as markers.
Other minor mistakes:
Line 47: compounds [8,9] – compounds [8,9].
Our response: We did this change.
Line 53: known as angico [14,15] – known as angico [14,15].
Our response: We did this change.
Line 81: 40° C. – 40°C.
Our response: We did this change.
In the all text: “proanthcynidins” – proanthocyanidins
Our response: We did this change.
Line 231: Purification – purification
Our response: We did this change.
Line 238: in in DPPH assay – in DPPH assay
Our response: We did this change.
Page 8, Table 1: Antimicrobian Activity – Antimicrobial Activity
Our response: We did this change.
Line 287: p-hydrobromic acid (1) – p-hydroxybenzoic acid (1)
Our response: We did this change.
Reviewer 2 Report
In the manuscript entitled “Bioguided purification of active compounds from leaves of Anadenanthera colubrina var. cebil (Griseb.) Altschul” the authors were performed a bioguided study to obtain compounds from A. colubrina var cebil with antimicrobial and antioxidant activity using the bioguide approach. The work is overall well done, carefully thought and performed and the manuscript is well written and easy to read and follow. All experimental methods are well explained. Other Specific comments:
The chemical characterizations have been well described.
Which the differences between Staphylococcus aureus ATCC 02 and S. aureus UFPEDA 705? Why use similar bacteria?
The results, and especially the discussion sections, do not provide effective indications of the mechanism of action or possible mechanism. This discussion topic does not use the chemical profile for demonstrations of antibacterial activity. I suggest promoting correlation of antioxidant and antimicrobial activity.
The quality of English writing throughout the manuscript is inferior, it is not necessary to list all errors and unprofessional expressions. Native or professional English writer assistance may be required
Author Response
Dear reviewer,
Reviewer 2
We would like to thank you for all suggestions that improved the overall quality of our manuscript. Following we provide detailed answers for each point.
Reviewer comment: Which the differences between Staphylococcus aureus ATCC 02 and S. aureus UFPEDA 705? Why use similar bacteria?
Our response: S. aureus UFPEDA 705 is a methicillin-resistant isolate obtained from surgical wound, and resistant to the following antibiotics: ampicillin, oxacillin, cephalothin, cefoxitin, cefepime, cefuroxime, nalidixic acid, nitrofurantoin and gentamicin
Reviewer comment: The results, and especially the discussion sections, do not provide effective indications of the mechanism of action or possible mechanism. This discussion topic does not use the chemical profile for demonstrations of antibacterial activity. I suggest promoting correlation of antioxidant and antimicrobial activity.
Our response: Although the antimicrobial action of the compounds obtained has been described in the literature, the action mechanisms have not been elucidated yet. We have added in the discussion more details about the action of these compounds, including the other medicinal properties attributed for each compound. In addition, we also added a description of the participation of oxidative stress in the pathogenesis of several clinical conditions.
Reviewer comment: The quality of English writing throughout the manuscript is inferior, it is not necessary to list all errors and unprofessional expressions. Native or professional English writer assistance may be required
Our response: We have corrected the grammar and orthographic mistake throughout the text.

Reviewer 3 Report
2 sep 2019.
Revisión de Anadenthera colubrine (Biomolecules)
CONSIDERATIONS.
The authors point out the importance of the use of medicinal plants in popular culture but also the fact that the composition of the plants is different (which depends on several factors) and therefore of the differences in their effects so it is important to identify the active components and quantities needed to achieve their effects.
They focus this study on the Anadenanthera colubrina leaves to which they undergo processes of extraction and fractionation and identification of compounds by chromatography, mass spectroscopy and nuclear magnetic resonance. In addition, the compounds were analyzed for antimicrobial (for Staphylococcus aureus ATCC 02 and S. aureus UFPEDA 705) and antioxidant capacities (DPPH, TAA and FRAP assays).
Among the numerous fractions obtained from acetonitrile/ethanol extract, five fractions were selected for analysis. The composition analysis showed the presence of quercetin, catechin, p-coumaric acid, gallic acid and mainly hyperosid, a compound that was obtained and identified for first time in leaves of A. colubrina.
There are some reports from A. colubrine, which used fruits (da Silva LC et al. Anti-Staphylococcus aureus action of three Caatinga fruits evaluated by electron microscopy. Nat Prod Res. 2013;27(16):1492-6; Agostini VO et al. Natural and non-toxic products from Fabaceae Brazilian plants as a replacement for traditional antifouling biocides: an inhibition potential against initial biofouling. Environ Sci Pollut Res Int. 2019 Jul 17), stem bark (Barreto HM et al. Enhancement of the antibiotic activity of aminoglycosides by extracts from Anadenanthera colubrine (Vell.) Brenan var. cebil against multi-drug resistant bacteria. Nat Prod Res. 2016;30(11):1289-92) but not leaves.
Also there are other 7 reports about Anadenanthera colubrina leaves but not related to antibacterial or antioxidant activities.
This is a well-designed and performed study. The conclusions are supported by the data presented.
Comments
line 210 Please clarify the significance value of P; is it 0.005% or 0.05%?
Table 1, correct antimicrobian. Please clarify dispersion, is it SD or SEM? How many replicates were made?
DPPH assay. Please clarify this phrase: “250 μL of 1 mM DPPH solution” given that the absorbance at 517 nm of 100 μM DPPH solution is near of 1.2. Which was the absorbance value of control tube?
Trolox is usually used as standard in DPPH assay, however its IC50 is not indicated, Why?
Minor points.
Page 3
line 98 Acetonitrile in lowercase
line 113 correct Mili-Q
line 122 10 μL of the sample was injected (were)
line 123 was maintained for 0.25 minutes (express as seconds)
line 124 The proportion of B was (refers as “phase B”)
line 134 was equipped with a 5mm multinuclear (separate “5mm”)
line144 by the Culture collection of the Department (write with uppercase)
line 150 in 10% DMSO (Dimethyl sulfoxide in distilled water) (first write the name and after the abbreviation)
line 158 add parentheses to equation to separate the multiplication by 100 (as in equation for TAA)
line 170 An aliquot of 250 μL of the DPPH methanolic solution (1 mM) (250 μL of DPPH methanolic)
line 183 concentration of 1mg/mL in methanol (separate “1mg/ml”)
line 184 1mL (separate)
line 185 4mM (separate)
line 195 Use the FRAP abbreviation
line 197 Eliminate the recipe for FRAP solution and indicate the concentration (and pH of buffer) or each component of FRAP reagent
line 198 for TPTZ (first write the name and after the abbreviation)
line 225 correct “based on analyzes of mass spectrometry”
Figure 1 legend, write Chromatogram with lowercase and include full stop.
line 231 use lowercase for Purification
Figure 2 legend, write Chromatogram with lowercase. Correct “chromatogram 3D”.
line 267, correct “groups positions was confirmed”
line 274 correct Hyperoside
lines 280, 281 siglets or singlets?
line 282 correct “uint”
line 284 add a full stop
line 290 Proanthcyanidin, use lowercase
line 307, 341 correct “bio-guided”
line 320 Improve the writing: “Following the anti-S. aureus activity, two fractions were selected to compounds purification, leading to obtention of four isolated compounds.”
line 323 improve the writing: “being the proanthocyanidins and p-hydroxybenzoic acid major components of the F2 and hyperoside and other major quercetin derivatives found in F4.”
lines 336-339 improve the writing: “These compounds are indicated as useful for the prevention and treatment of various diseases related to oxidative stress such a neurodegenerative and cardiovascular and are more accessible and without side effects compared to synthetic compounds.”
Author Response
Reviewer 3
Dear reviewer,
We would like to thank you for all suggestions that improved the overall quality of our manuscript. Following we provide detailed answers for each point.
Reviewer comment: line 210 Please clarify the significance value of P; is it 0.005% or 0.05%?
Our response: We corrected for 0.05
Reviewer comment: Table 1, correct antimicrobian. Please clarify dispersion, is it SD or SEM? How many replicates were made?
Our response: We corrected this mistake and added more information on statistical analysis.
Reviewer comment: DPPH assay. Please clarify this phrase: “250 μL of 1 mM DPPH solution” given that the absorbance at 517 nm of 100 μM DPPH solution is near of 1.2. Which was the absorbance value of control tube?
Our response: We added the solution absorbance (0.7) in the section.
Reviewer comment: Trolox is usually used as standard in DPPH assay, however its IC50 is not indicated, Why?
Our response: For DPPH assay we used gallic acid as control.
All the minors correctios suggested by the reviewer 3 were performed and are highlighted in yellow.

Reviewer 4 Report
this is excellent work related to the use of natural compounds as antimicrobial.
I only have few comments:
can you provide limitations to the study in the conclusion, what are possible limitations of this work? what is the future directions? thank you
Author Response
Dear reviewer,
We would like to thank you for your attention. We have added in the conclusion the points that were not addressed in our work and indicated the directions of future studies using these compounds.

Round 2
Reviewer 1 Report
Dear Authors,
The manuscript ID: biomolecules-593126-v2 entitled “Bioguided purification of active compounds from leaves of Anadenanthera colubrina var. cebil (Griseb.) Altschul” has been partially corrected.
I still have some suggestions in order to improve paper, which are the following:
Line 24: In the whole text is MIC50 (dose able to inhibit 50% of bacteria growth) ????
Determination of the MIC values is incomprehensible. MIC50 value is important parameter for reporting results of susceptibility testing when multiple isolates of a given species are tested. MIC50 value is defined as the lowest concentration of the compound at which 50% of these isolates were inhibited. The MIC50 requires testing about (usually at least) 100 isolates, gives the MIC, which inhibits 50% of these isolates. As an example, in a test population of 70 strains, the MIC50 is the value at position 35 in a graded series of MICs starting with the lowest MIC value at position 1. In turn, in a test population of 71 strains, the MIC50 is the value at position 36 in the aforementioned graded series of the MICs [Schwarz et al., 2010].
[Stefan Schwarz, Peter Silley, Shabbir Simjee, Neil Woodford, Engeline van Duijkeren, Alan P. Johnson, Wim Gaastra: Editorial: Assessing the antimicrobial susceptibility of bacteria obtained from animals..Journal of Antimicrobial Chemotherapy, Volume 65, Issue 4, April 2010, Pages 601–604, https://doi.org/10.1093/jac/dkq037]
Perhaps, it is here the value of IC50 – half-maximal inhibitory concentration ????
In microbiology, the minimum inhibitory concentration – MIC is the lowest concentration of drugs, compounds, extracts or other substances, which prevents visible growth of microorganisms.
Moreover, according to Quave at al., 2008 [28], this is the average% inhibition of repeated tests used to determine the final MIC values:
% inhibition = [1 - (ODT24h-ODT0h) / (ODgc24h-ODgc0h)] x 100
In the text – Line 166 is:
% inhibition = [(ODT24h-DOT0h) / (ODgc24h-ODgc0h)] x 100
Please correct this equation and eventually results.
Line 147: Please add: S. aureus UFPEDA 705 is a methicillin-resistant (MRSA – Methicillin Resistant S. aureus) isolate
Lines 148-149: Please add: resistant to the following antibiotics including drugs from the beta-lactam group (e.g. ampicillin, oxacillin, cephalothin, cefoxitin, cefepime, cefuroxime) and other nalidixic acid, nitrofurantoin and gentamicin [27].
Other minor mistakes:
In the whole text, please correct: proanthcynidin and/or proanthcyanidin – proanthocyanidin
Line 158: extrac – extract
With highest regards,
Author Response
Dear reviewer,
We would like to thank you again for all your suggestions.
We have correct the equation for calculation of antimicrobial action and change the designation for IN50 instead of MIC50.
We also changed all minor points and these alterations are highlighted in green.
With highest regards,
